# CP100356 Hydrochloride, a P-Glycoprotein Inhibitor, Inhibits Lassa Virus Entry: Implication of a Candidate Pan-Mammarenavirus Entry Inhibitor

**DOI:** 10.3390/v13091763

**Published:** 2021-09-03

**Authors:** Toru Takenaga, Zihan Zhang, Yukiko Muramoto, Sarah Katharina Fehling, Ai Hirabayashi, Yuki Takamatsu, Junichi Kajikawa, Sho Miyamoto, Masahiro Nakano, Shuzo Urata, Allison Groseth, Thomas Strecker, Takeshi Noda

**Affiliations:** 1Laboratory of Ultrastructural Virology, Institute for Frontier Life and Medical Sciences, Kyoto University, Shogoin-Kawahara-cho 53, Sakyo-ku, Kyoto 606-8507, Japan; takenaga.toru.4z@kyoto-u.ac.jp (T.T.); zhang.zihan.57w@st.kyoto-u.ac.jp (Z.Z.); muramoto.yukiko.5e@kyoto-u.ac.jp (Y.M.); hirabayashi.ai.2s@kyoto-u.ac.jp (A.H.); yukiti@niid.go.jp (Y.T.); kajikawa.junichi.74m@st.kyoto-u.ac.jp (J.K.); s-miyamo@nih.go.jp (S.M.); nakano.masahiro.4z@kyoto-u.ac.jp (M.N.); 2Laboratory of Ultrastructural Virology, Graduate School of Biostudies, Kyoto University, 53 Shogoin Kawahara-cho, Sakyo-ku, Kyoto 606-8507, Japan; 3CREST, Japan Science and Technology Agency, 4-1-8 Honcho, Kawaguchi, Saitama 332-0012, Japan; 4Institute of Virology, Phillips University Marburg, Hans-Meerwein-Strasse 2, 35043 Marburg, Germany; fehling@staff.uni-marburg.de (S.K.F.); strecker@staff.uni-marburg.de (T.S.); 5National Research Center for the Control and Prevention of Infectious Diseases (CCPID), Nagasaki University, 1-12-4 Sakamoto, Nagasaki 852-8523, Japan; shuzourata@nagasaki-u.ac.jp; 6Laboratory for Arenavirus Biology, Institute of Molecular Virology and Cell Biology, Friedrich-Loeffler-Institut, Südufer 10, 17493 Greifswald-Insel Riems, Germany; Allison.Groseth@fli.de

**Keywords:** Lassa virus, lymphocytic choriomeningitis virus, arenavirus, pseudotyped vesicular stomatitis virus, entry inhibitor

## Abstract

Lassa virus (LASV)—a member of the family *Arenaviridae*—causes Lassa fever in humans and is endemic in West Africa. Currently, no approved drugs are available. We screened 2480 small compounds for their potential antiviral activity using pseudotyped vesicular stomatitis virus harboring the LASV glycoprotein (VSV-LASVGP) and a related prototypic arenavirus, lymphocytic choriomeningitis virus (LCMV). Follow-up studies confirmed that CP100356 hydrochloride (CP100356), a specific P-glycoprotein (P-gp) inhibitor, suppressed VSV-LASVGP, LCMV, and LASV infection with half maximal inhibitory concentrations of 0.52, 0.54, and 0.062 μM, respectively, without significant cytotoxicity. Although CP100356 did not block receptor binding at the cell surface, it inhibited low-pH-dependent membrane fusion mediated by arenavirus glycoproteins. P-gp downregulation did not cause a significant reduction in either VSV-LASVGP or LCMV infection, suggesting that P-gp itself is unlikely to be involved in arenavirus entry. Finally, our data also indicate that CP100356 inhibits the infection by other mammarenaviruses. Thus, our findings suggest that CP100356 can be considered as an effective virus entry inhibitor for LASV and other highly pathogenic mammarenaviruses.

## 1. Introduction

Viruses belonging to the *Arenaviridae* family that infect mammals (mammarenaviruses) are serologically and geographically classified into the Old World and New World mammarenaviruses. The Old World serogroup includes Lassa virus (LASV), the causative agent of Lassa fever in West Africa, which in its severe forms results in hemorrhagic fever and lymphocytic choriomeningitis virus (LCMV), which in severe cases causes neurological disease in humans [1,2]. New World viruses, i.e., Junín, Machupo, Guanarito, Sabiá, and Chapare viruses, cause hemorrhagic fever and neurological disease in humans in South America [3]. These highly pathogenic mammarenaviruses including LASV are classified as Category A bioterrorism agents and must be handled in a biosafety level-4 (BSL-4) facility. Among mammarenaviruses, the most devastating with respect to its impact on public health is LASV, with more than 300,000 infections and approximately 5000 deaths estimated per year in West Africa [3,4]. No approved antiviral drugs or vaccines are currently available for the treatment or prevention of LASV infection. Ribavirin, a ribosyl purine analogue, is commonly used off-label to treat Lassa fever patients, although recent studies increasingly question the extent of its effectiveness when used alone [5,6,7,8,9]. Several other anti-LASV compounds are under investigation, such as favipiravir, a purine analogue that inhibits viral transcription and replication [8]; ST-193, a benzimidazole derivative that inhibits viral entry [10]; PF-429242, a site 1 protease inhibitor that blocks infectious LASV formation [11,12]; isavuconazole, an antifungal agent that inhibits membrane fusion [13]; losmapimod, a p38 mitogen-activated protein kinase inhibitor that blocks membrane fusion [14]; and lacidipine, a lipophilic dihydropyridine calcium antagonist that inhibits membrane fusion [15]. However, there is so far only limited data demonstrating their therapeutic effectiveness against LASV.

LASV consists of enveloped pleomorphic particles with a diameter of 100–300 nm, in which their bi-segmented ambisense RNA genome is incorporated. The large RNA segment encodes the RNA-dependent RNA polymerase L and the matrix protein Z, while the small RNA segment encodes the nucleoprotein NP and the surface glycoprotein complex GP. The LASV glycoprotein (LASVGP) undergoes proteolytic cleavage by signal peptidase and by proprotein-convertase subtilisin kexin isozyme-1/site 1 protease in the ER and Golgi apparatus to yield an SSP/GP1/GP2 complex that is then incorporated as heterotrimers into the LASV virion envelope [16,17,18,19,20]. The GP1 subunit binds to the cellular receptor α-dystroglycan (α-DG) [21], whereas GP2 is responsible for membrane fusion after receptor-mediated endocytosis, which depends on the interaction of GP1 with late endosomal proteins such as lysosome-associated membrane protein 1 [22].

Studies using authentic LASV must be performed in a BSL-4 laboratory, and this restriction delays the research and development of anti-LASV drugs and, in particular, tends to limit access to the specialized equipment needed for high-throughput antiviral testing. However, since LASV entry is mediated solely by LASVGP, use of a pseudotyped virus system harboring LASVGP is a reasonable option for the initial phases of antiviral screening for LASV entry inhibitors. In addition, because the prototype Old World arenavirus LCMV is relatively low pathogenic, and thus suitable for use in BSL-2 facilities; it can be also be useful for screening to identify candidate inhibitors for LASV. Hence, in this study, we screened a chemical library consisting of 2480 small compounds to identify potential LASV entry inhibitors using a pseudotyped vesicular stomatitis virus (VSV) harboring LASVGP (VSV-LASVGP), as well as wild-type VSV as a control, and the prototypic LCMV. After identification of a compound that inhibits both VSV-LASVGP and LCMV replication (i.e., CP100356 hydrochloride), we confirmed its antiviral activity using authentic LASV. Further, we determined that the compound also inhibits VSV pseudotypes containing GP from other mammarenaviruses from the New World serogroups. Since the compound acts on P-glycoprotein (P-gp), which is an integral membrane protein in the plasma membrane and is an ATP-binding cassette transporter that exports various chemicals out of the cells [23,24], we also investigated the impact of P-gp downregulation on the virus replication. 

## 2. Materials and Methods

### 2.1. Cells, Viruses, Plasmids, and Compounds 

Vero cells and Vero E6 cells were cultured in Eagle’s minimum essential medium (MEM) supplemented with 10% fetal calf serum, 1% non-essential amino acid solution, 1 mM sodium pyruvate, 5 mM L-glutamine, 50 U/mL penicillin, and 50 μg/mL streptomycin at 37 °C with 5% CO_2_. BHK-21 cells were cultured in MEM supplemented with 1 mM sodium pyruvate at 37 °C with 5% CO_2_. HEK293Τ cells were cultured as for the BHK-21 cells, but using Dulbecco’s Modified Eagle’s medium (DMEM). The replication-competent VSV pseudotypes expressing LASVGP from either strain Josiah or Alzey were prepared as previously described [25], while the wild-type VSV was propagated in Vero cells. LCMV (strain Armstrong 53b) was propagated in BHK-21 cells. Authentic LASV (strain Josiah) was propagated in Vero E6 cells. For pseudotyping, the LASVGP genes from the Josiah strain (GenBank. NC_004296.1) or the Alzey strain (GenBank. LT601602.1), which was isolated from the first case of acquired Lassa fever outside of Africa [26]; the LCMVGP gene from the Armstrong 53b strain (GenBank. AY847350); the Junín virus GP (JUNVGP) gene from the Romero strain (GenBank. AY619641.1); the Tacaribe virus GP (TCRVGP) gene from the TRVL-11573 strain (GenBank. MT081316.1); the Machupo virus GP (MACVGP) gene from the Chicava strain (GenBank. NC_005078.1); and the Sabia virus GP (SABVGP) gene from the SPH114202 strain (GenBank. NC_006317.1) were cloned into the mammalian expression vector pCAGGS. The vesicular stomatitis virus G glycoprotein (VSVG) gene from the Indiana strain (GenBank. NC_001560.1) was cloned into the mammalian expression vector pCMV. For screening, a total of 2480 small compounds were provided by the Medical Research Support Center, Graduate School of Medicine, Kyoto University. For further experiments, CP100356 (CAS RN. 142715-48-8), lacidipine (CAS RN. 103890-78-4) and Tariquidar (CAS RN. 206873-63-4) were purchased from Sigma-Aldrich (Saint Louis, MO, USA), Tokyo Chemical Industry (Tokyo, Japan), and Cayman Chemical (Ann Arbor, MI, USA), respectively. 

### 2.2. Screening Based on Cell Survival

Vero cells grown in 96-well plates were infected with VSV-LASVGP at a multiplicity of infection (MOI) of 0.01, and treated with the respective compounds at a final concentration of 10 μM for 22 h. The proportion of surviving cells left in the plates, which represent those cells resistant to VSV-LASVGP replication, were quantified automatically using the Cytell image cytometer (GE Healthcare, Chicago, IL, USA).

### 2.3. Evaluation of Antiviral Activity

For VSV-LASVGP and VSV infection, Vero cells were pre-treated with the respective compounds at the indicated concentrations for 1 h, infected with the indicated virus at an MOI of 0.01, and cultured in the presence of the respective compounds for a further 24 h. Virus titers were determined by performing the plaque assay using Vero cells. For LCMV infection, Vero cells were pre-treated for 1 h with the respective compounds at the indicated concentrations, infected with the virus at an MOI of 0.01, in the presence of the respective compounds, and cultured in the presence of the compounds for a further 48 h. Virus titers were determined by immunofluorescence assay using Vero cells and an anti-LCMV NP rat antibody (VL-4, BioXCell). 

### 2.4. Validation of Antiviral Activity

For infection with authentic LASV, Vero E6 cells were pre-treated for 1 h with the respective compounds at the indicated concentrations, infected with the virus at an MOI of 0.01 in the presence of the respective compounds, and cultured in the presence of the respective compounds for a further 24 or 48 h. Virus titers were determined by performing a median tissue culture infectious dose (TCID_50_) assay, as described in 2.5. All the experiments with infectious LASV were performed in the BSL-4 facility at the Institut für Virologie, Philipps-Universität Marburg, Marburg, Germany.

### 2.5. TCID_50_ Assay

Samples containing authentic LASV were titrated using the TCID_50_ assay, as described previously [27] using Vero E6 cells, and the observations were recorded at 7 days post-infection (dpi). The virus titers were determined using the Reed–Muench method [28]. 

### 2.6. Cell Viability Assay

To evaluate cell viability, cells were collected at 24 or 48 h after compound treatment, and the amount of intracellular ATP was measured by performing the CellTiter-Glo assay according to the manufacturer’s protocol (Promega, Madison, WI, USA).

### 2.7. Receptor Binding Assay with Quantitative Reverse Transcription Polymerase Chain Reaction (RT-qPCR)

Vero cells were pre-treated with either a vehicle control (dimethyl sulfoxide; DMSO) or 20 µM CP100356 for 1 h. VSV-LASVGP at an MOI 460 was adsorbed onto the cells in the presence of the respective compounds at 4 °C for 1 h. After washing with phosphate-buffered saline (PBS), total RNA was collected from the cells using an RNeasy kit (QIAGEN, Hilden, Germany). RT-qPCR was performed using random hexamers (Thermo Fisher Scientific, Waltham, MA, USA) and specific primers for the detection of the VSV L gene, according to a previous report [29]. The results were normalized against the expression of β-actin mRNA. 

### 2.8. Virus Fusion Assay

The assay was performed as described previously with some modifications [15]. HEK293Τ cells grown in 24-well plates were transfected with plasmids expressing either LASVGP, LCMVGP, or VSVG, together with a plasmid expressing green fluorescent protein (GFP), and cultured for 24 h (for LASVGP and LCMVGP) or 8 h (for VSVG). After 1 h of treatment with 10 μM of the respective compounds, the medium was replaced with PBS (pH 4.5) containing the compound and incubated for 10 min [22,30]. The PBS (pH 4.5) was then replaced with cell culture media, and syncytium formation was observed using a fluorescence microscope at 1.5–2 h after the low-pH treatment. 

### 2.9. Electron Microscopy

Vero cells were pre-treated with CP100356 or a vehicle control (DMSO) for 1 h. VSV-LASVGP or VSV at an MOI of 460 was adsorbed onto the cells at 4 °C for 1 h. The cells were then incubated at 37 °C for 30 min, and fixed with 2.5% glutaraldehyde in 0.1 M cacodylate buffer. After fixation with 2% osmium tetroxide, the cells were dehydrated with a series of ethanol gradients, followed by propylene oxide, and embedded in Epon 812 resin (TAAB). The thin sections were stained with uranyl acetate and lead citrate, and observed using a Hitachi HT-7700 at 80 kV.

### 2.10. Downregulation of P-Glycoprotein (P-gp) by Small Interfering RNA (siRNA)

Vero cells cultured in antibiotic-free 10% FCS/DMEM in 24-well plates were transfected with 1.25 pmol of the Silencer Select Negative Control no. 1 siRNA (Thermo Fisher Scientific, Waltham, MA, USA) or the Silencer Select Predesigned human ATP-binding cassette subfamily B member 1 siRNA (ID: s10418, Thermo Fisher Scientific, Waltham, MA, USA) targeting P-gp, using Lipofectamine RNAiMAX transfection reagent (Thermo Fisher Scientific, Waltham, MA, USA). A second transfection was performed 24 h after the first transfection. Forty-eight hours after the second transfection, the Vero cells were infected with 1000 plaque-forming units of VSV-LASVGP or VSV, or 1000 focus forming units of LCMV, and cultured for 24 h (for VSV-LASVGP and VSV) or 48 h (for LCMV). The supernatant was collected for viral titration. Western blotting was performed using the anti-P-glycoprotein antibody (ab170904, Abcam) and anti-α-tubulin antibody (PM054, MBL).

### 2.11. Statistical Analysis

Excel (Microsoft) or Prism 9 (GraphPad Software, San Diego, CA, USA) was used to generate graphs. The sample size varied per experiment and is indicated in each figure legend. We compared group means by the Welch *t*-test, and compared each group with the indicated control using Prism 9. All the statistical tests were two-sided. A *p*-value of <0.05 was considered statistically significant.

## 3. Results

### 3.1. Screening of Small Compounds That Inhibit VSV-LASVGP and LCMV Infection

To identify potential LASV entry inhibitors, we screened 2480 compounds using VSV-LASVGP. The library of the Medical Research Support Center, Kyoto University, comprises pharmacologically active and validated small compounds that are commercially available. Vero cells grown in 96-well plates were treated with each of the respective compounds at a concentration of 10 μM and were infected with VSV-LASVGP at an MOI of 0.01. At 22 h post-infection (hpi), the proportion of surviving cells, indicating resistance to VSV-LASVGP infection, were quantified automatically using an image cytometer. The Vero cells treated with a vehicle control (0.1% DMSO) or 30 μM lacidipine, a validated LASV entry inhibitor [15], were also included on each plate to allow calculation of a Z’ score, indicating consistency in assay performance. The Z’ score was more than 0.5 for all the plates, indicating a high consistency in our screening system [31]. Through this primary screening, we identified 78 hits for compounds showing higher cell survival rates than for 30 μM lacidipine after VSV-LASVGP infection (Figure 1A). Next, to determine whether these primary hit compounds actually suppressed VSV-LASVGP infection, Vero cells were treated with the respective primary hit compounds at a concentration of 10 μM, infected with VSV-LASVGP at an MOI of 0.01, and the virus titers in the cell culture supernatants were determined at 24 hpi. Among the 78 primary hit compounds, 10 hit compounds showed more than a 10-fold reduction in VSV-LASVGP replication after treatment at a 10 μM concentration (Figure 1A). To exclude compounds targeting VSV replication, independent of LASVGP, Vero cells were pre-treated with the 10 remaining hit compounds at a 10 μM concentration and were infected with VSV at an MOI of 0.01, and the virus titers were examined at 24 hpi. Out of the 10 compounds, nine compounds showed less than a two-fold reduction in VSV titers compared with the vehicle control (Figure 1A), suggesting that these nine compounds specifically suppressed LASVGP-mediated virus infection.

It is considered that LCMV, a prototype Old World arenavirus, employs a similar entry mechanism and can be used as a surrogate for LASV infection under BSL-2 conditions. Therefore, to investigate whether these nine hit compounds suppressed LCMV replication, Vero cells pre-treated with the remaining nine hit compounds at 10 μM concentration were infected with LCMV at an MOI of 0.01, and the virus titers were determined at 48 hpi. Four out of the nine hit compounds showed more than a 1000-fold reduction in LCMV titers (Figure 1A). Two out of the four compounds showed significant cytotoxicity at 10 μM concentration, as indicated by a cell viability rate of less than 90%. Among the two compounds without significant cytotoxicity (i.e., CP100356 hydrochloride [4-(3,4-Dihydro-6,7-dimethoxy-2(1H)-isoquinolinyl)-N-2[2-(3,4-dimethoxyphenyl)ethyl]-6,7-dimethoxy-2-quinazolinamine hydrochloride] and MCOPPB trihydrochloride hydrate [1-[1-(1-Methylcyclooctyl)-4-piperidinyl]-2-(3R)-3-piperidinyl-1H-benzimidazoletrihydrochloride]), CP100356 hydrochloride (hereafter CP100356) was selected for further investigation (Figure 1B), since its target P-gp is highly expressed on epithelial cell surfaces in the liver, kidney, and gastrointestinal tract, some of which are important replication sites of LASV in vivo [32].

### 3.2. Antiviral Effect of CP100356 on VSV-LASVGP, LCMV, and Authentic LASV

We first evaluated the 50% inhibitory concentration (IC_50_) and 50% cytotoxic concentration (CC_50_) of CP100356. Vero cells treated with CP100356 at a starting concentration of 50 μM with three-fold serial dilutions were infected with either VSV-LASVGP, LCMV, or VSV. The supernatants were collected at indicated time points, and the respective virus titers were determined. CP100356 inhibited the replication of VSV-LASVGP and LCMV with IC_50_ values of 0.52 and 0.54 μM, respectively, and it did not exhibit significant cytotoxicity at these concentrations, with a CC_50_ of 21.0 μM (Figure 2A–C). Although CP100356 exhibited modest suppression of VSV replication at an IC_50_ of 3.6 μM (Figure 2D), this value was approximately seven-fold higher than that for VSV-LASVGP and LCMV. 

Next, to assess the antiviral activity of CP100356 on authentic LASV, infection experiments were performed under BSL-4 conditions. Vero E6 cells pre-treated with different concentrations of CP100356 were infected with LASV at an MOI of 0.01, and cultured in the presence of this compound for 2 days. Then, the virus titers were determined by TCID_50_ assay. CP100356 efficiently inhibited LASV replication in a dose-dependent manner (Figure 2E,F), with concentrations above 1.85 μM showing a significant reduction in the virus titers at 1 and 2 dpi. For example, treatment with 5.56 μM CP1000356 reduced the virus titers to 0.1% and 0.001% compared with mock treatment at 1 and 2 dpi, respectively. The IC_50_ value was 0.062 μM based on the virus titers at 2 dpi. Taken together, these results indicate the efficient antiviral activity of CP100356 for authentic LASV as well as LCMV.

### 3.3. Inhibition of Viral Membrane Fusion by CP100356

Entry of LASV into target cells starts with GP-mediated receptor binding on the cell surface. To test whether CP100356 blocks receptor binding, VSV-LASVGP was adsorbed on ice onto Vero cells pre-treated with CP100356. After extensive washing with PBS, viral RNA was extracted from the virus attached to the cells and subjected to RT-qPCR using a primer set targeting the VSV L gene. No significant difference was found in the amount of viral RNA extracted from bound particles between CP100356 and vehicle control-treated cells (Figure 3A), suggesting that CP100356 does not competitively block receptor binding of LASVGP. 

Next, to investigate whether low-pH induced membrane fusion was blocked by CP100356, a cell-based membrane fusion assay was performed. HEK293Τ cells, which were transfected with a plasmid expressing either LASVGP, LCMVGP, or VSVG together with a plasmid expressing GFP, were treated with 10 μM CP100356, and exposed to a low-pH environment to induce cell-to-cell membrane fusion. In the absence of CP100356, syncytium formation, which is visible as a darker amorphous GFP signal, was easily observed in all the samples after low-pH stimulation but not in the samples without low-pH stimulation (Figure 3B). This confirmed that these viral proteins can induce membrane fusion in a low pH-dependent manner. Under these conditions, cell-to-cell membrane fusion was blocked in the cells expressing LASVGP and LCMVGP in the presence of CP100356, but no effect was observed in the cells expressing VSVG (Figure 3B). Next, to investigate whether CP100356 inhibits LASVGP-mediated membrane fusion in the context of virus entry, the Vero cells treated with 20 μM CP100356 were infected with either VSV-LASVGP or VSV, and were subject to ultrathin-section electron microscopy. At 30 min after infection, the VSV-LASVGP-infected cells treated with CP100356 showed an accumulation of virus particles in the intracellular vesicles, whereas those without CP100356 treatment did not (Figure 3C,D). In the VSV-infected cells, the apparent accumulation of virus particles in the intracellular vesicles was not observed after CP100356 treatment. Taken together, these results indicate that CP100356 inhibits LASVGP-mediated membrane fusion, resulting in the inhibition of LASV infection.

### 3.4. The Role of P-gp in LASVGP-Mediated Virus Entry

Although CP100356 inhibited VSV-LASVGP entry, the importance of its target, P-gp, for LASVGP-mediated virus entry, was unclear. P-gp was downregulated by siRNA, and its effect on virus replication was evaluated. In Vero cells transfected with the siRNA, the level of P-gp was substantially decreased 48 h after transfection (Figure 4A). The efficiency of VSV-LASVGP and VSV replication in the knockdown cells was comparable with that in the control cells (Figure 4B), suggesting that the inhibitory effect of CP100356 on LASVGP-mediated membrane fusion is independent of P-gp.

To further confirm that the antiviral effect of CP-100356 is independent of P-gp, another P-gp inhibitor, Tariquidar, was tested. Vero cells treated with 10 μM Tariquidar were infected with either VSV-LASVGP, LCMV, or VSV, and the respective virus titers were determined. Tariquidar did not inhibit the replication of all viruses tested without significant cytotoxicity (Figure 4C,D), suggesting that P-gp unlikely plays a role in LASVGP-mediated virus entry. 

### 3.5. Inhibitory Effect of CP100356 against Other Mammarenaviruses

Since CP100356 showed antiviral activity against LASV and LCMV, we further investigated whether it also inhibits the entry of other mammarenaviruses. We generated replication-incompetent pseudotyped VSVs harboring LASVGP (Josiah strain), LASVGP (Alzey strain), LCMVGP, Junín virus GP (JUNVGP), Machupo virus GP (MACVGP), Tacaribe virus GP (TCRVGP), or Sabiá virus GP (SABVGP), all of which express GFP in the infected cells. Consistent with the results obtained using replication-competent VSV-LASVGP, LCMV, and VSV (Figure 2B–D), treatment with 5.6 μM CP100356 significantly inhibited virus entry mediated by LASVGP and LCMVGP, whereas only a slight inhibition of VSVG-mediated virus entry was observed (Figure 5). Interestingly, CP100356 significantly inhibited the entry of all mammarenavirus GP-mediated viruses. Taken together, these results strongly suggest that CP100356 acts as a pan-mammarenavirus entry inhibitor.

## 4. Discussion

In this study, we screened 2480 small compounds using VSV-LASVGP and LCMV. CP100356, a P-gp inhibitor, was identified as a potential candidate drug against LASV, as well as other mammarenaviruses. CP100356 efficiently blocked low pH-dependent membrane fusion mediated by LASVGP and LCMVGP, thereby inhibiting VSV-LASVGP and LCMV infection. It also suppressed authentic LASV infection with an IC_50_ of 0.062 μM. Interestingly, it significantly inhibited the entry of pseudotyped viruses harboring various mammarenavirus GPs as well, including representatives of both the Old World and New World groups. Therefore, our results indicate that CP100356 is a promising candidate for the development of antiviral compounds against LASV and other mammarenaviruses that cause severe disease in humans. 

CP100356 specifically binds to the drug-binding site at the transmembrane region of P-gp and inhibits the efflux of various substrates [33,34]. Because P-gp overexpressing cells with increased efflux pump function also show resistance to several enveloped virus infections [35], inhibition of the efflux pump is unlikely responsible for the antiviral activity. Notably, P-gp downregulation as well as Tariquidar treatment exhibited no direct effect on virus replication (Figure 4B,D). These results suggest that neither the efflux pump function of P-gp, nor P-gp itself, is involved in the entry of LASV and LCMV. The exact mechanisms by which it blocks membrane fusion mediated by LASVGP and LCMVGP remains to be determined. Considering that CP100356 inhibits the GP-mediated entry of various mammarenaviruses, including those that are only distantly related and use different entry receptors, CP100356 also does not appear to directly act on arenavirus GPs, but on as yet-to-be identified host factors and/or the viral lipid envelope. Further studies are, therefore, needed to reveal the underlying molecular mechanism associated with the inhibition of membrane fusion by CP100356.

In conclusion, this is the first report of CP100356 being identified as a viral entry inhibitor. CP100356 effectively inhibits LASV entry, and thus represents a promising lead compound for the development of effective therapeutic strategies against the highly pathogenic LASV. Since it also inhibits pseudotyped virus entry mediated via the GP of various other mammarenaviruses, it is also expected to exert antiviral activity against highly pathogenic New World mammarenaviruses. 

## Figures and Tables

**Figure 1 viruses-13-01763-f001:**
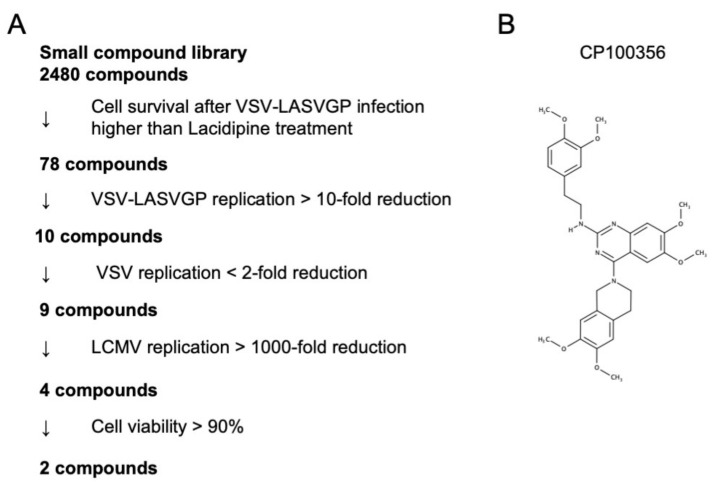
Screening for Lassa virus entry inhibitors from the drug library of Kyoto University. (**A**) Assay workflow. (**B**) Chemical structure of CP100356.

**Figure 2 viruses-13-01763-f002:**
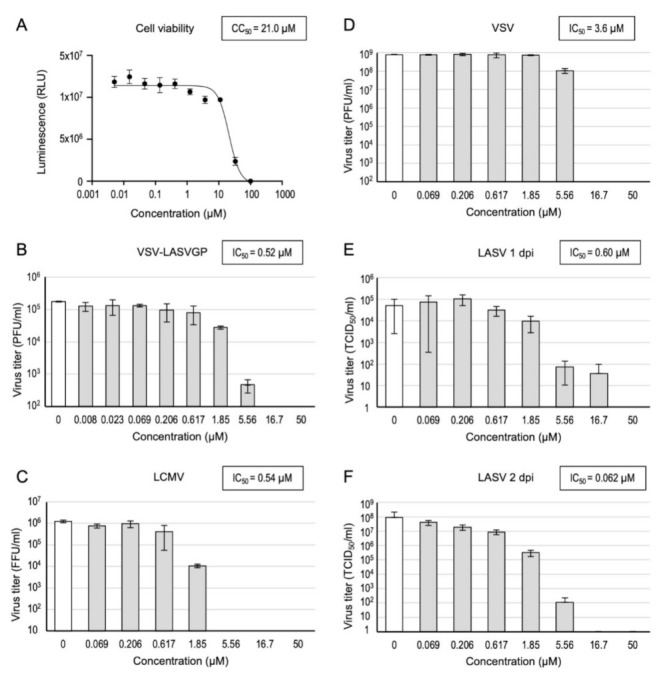
Antiviral effects of CP100356 on pseudotyped vesicular stomatitis virus harboring LASV glycoprotein (VSV-LASVGP), lymphocytic choriomeningitis virus (LCMV), and authentic LASV. (**A**) Cell viability of Vero cells was measured by performing a CellTiter-Glo assay after 24 h of treatment with CP100356 at the indicated concentrations. (**B**) Vero cells treated with the indicated concentrations of CP100356 were infected with VSV-LASVGP at a multiplicity of infection (MOI) of 0.01, and virus titers were determined by performing the plaque assay at 24 h post-infection (hpi). (**C**) Vero cells treated with the indicated concentrations of CP100356 were infected with LCMV at an MOI of 0.01, and virus titers were determined at 48 hpi by counting focus-forming units using an anti-NP antibody. (**D**) Vero cells treated with the indicated concentrations of CP100356 were infected with VSV at an MOI of 0.01, and virus titers were determined by plaque assay at 24 hpi. (**E**,**F**) Vero cells treated with the indicated concentrations of CP100356 were infected with LASV at an MOI of 0.01, and virus titers were determined at 1 day post-infection (dpi) (**E**) and 2 dpi (**F**) by median tissue culture infectious dose assay. The values indicate means ± standard deviation of the experiment in biological triplicates for each concentration of CP100356.

**Figure 3 viruses-13-01763-f003:**
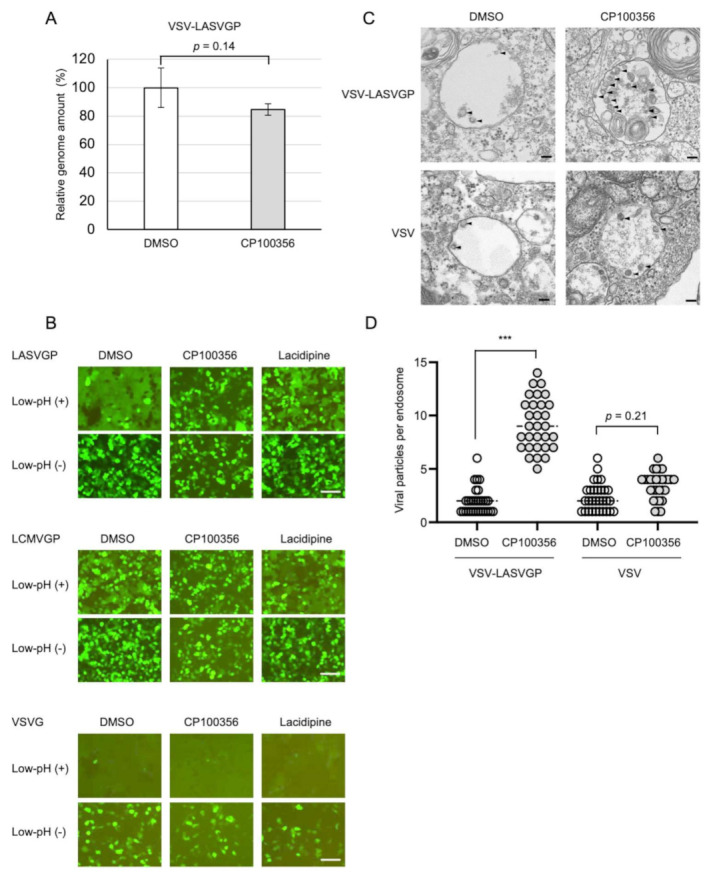
Mechanism of viral entry inhibition by CP100356. (**A**) Virus binding assay. Vero cells were pre-treated with the vehicle control (DMSO) or 20 µM CP100356 for 1 h. Pseudotyped vesicular stomatitis virus harboring Lassa virus glycoprotein (VSV-LASVGP) was adsorbed onto the cells at 4 °C in the presence of the respective compounds, and total RNA was extracted from the virus bound to the cells. Then quantitative reverse transcription polymerase chain reaction (RT-qPCR) was performed using the primers for VSV L, to evaluated virus attachment was. Welch’s t-test was used to compare the two groups and *p* values are indicated. The values indicate means ± standard deviation of the representative experiment in biological triplicates for each concentration of CP100356 from two independent experiments. (**B**) Cell-based membrane fusion assay. HEK293T cells expressing LASVGP, lymphocytic choriomeningitis virus glycoprotein (LCMVGP), or vesicular stomatitis virus glycoprotein (VSVG) were treated with 10 μM CP100356, 25 μM lacidipine, or the vehicle control (DMSO), and were subject to low pH (pH 4.5) treatment for 10 min. Syncytium formation was visualized by fluorescent microcopy. The images are representative fields from two independent experiments. Bars, 500 µm. (**C**) Vero cells treated with the vehicle control (DMSO) or 20 μM CP100356, and infected with either VSV-LASVGP or VSV, were subject to ultrathin-section electron microscopy at 30 min after the infection. The arrowheads show virus particles in the endosome. Bars, 100 nm. (**D**) Quantification of endosomal viral particles. Dots represent the number of viral particles observed in randomly selected electron microscopic images of endosomes (*n* = 30). Welch’s t-test was used to compare the two groups and *p* values are indicated; *** *p* < 0.001.

**Figure 4 viruses-13-01763-f004:**
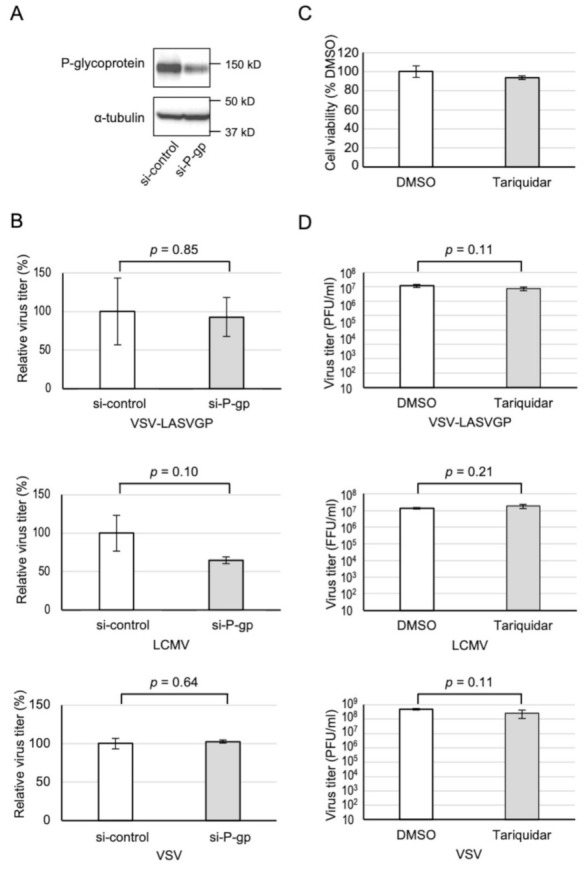
Effects of P-gp knockdown and another P-gp inhibitor on virus replication. (**A**) Vero cells were transfected with a control siRNA or an siRNA against P-gp, and the cell lysates were collected for Western blotting at 24 h post-infection (hpi). (**B**) Vero cells were transfected with a control siRNA or an siRNA against P-gp, followed by infection with 1000 plaque-forming units (PFU) of pseudotyped vesicular stomatitis virus harboring LASV glycoprotein (VSV-LASVGP), 1000 focus-forming units of lymphocytic choriomeningitis virus (LCMV), or 1000 PFU of VSV. At 24 hpi (VSV-LASVGP and VSV) or 48 hpi (LCMV), the supernatants were collected for virus titration. Welch’s t-test was used to compare the two groups and *p* values are indicated. The values indicate means ± standard deviation of the representative experiment in biological triplicates from two independent experiments. (**C**,**D**) Vero cells treated with 10 μM Tariquidar were infected with either VSV-LASVGP, LCMV, or VSV at a multiplicity of infection (MOI) of 0.01. At 24 hpi (VSV-LASVGP and VSV) or 48 hpi (LCMV), the supernatants were collected for virus titration. (**C**) Cell viability was measured by a CellTiter-Glo assay after 24 h of treatment with 10 μM Tariquidar. Welch’s t-test was used to compare the two groups and *p* values are indicated. The values indicate means ± standard deviation of the representative experiment in biological triplicates from three independent experiments.

**Figure 5 viruses-13-01763-f005:**
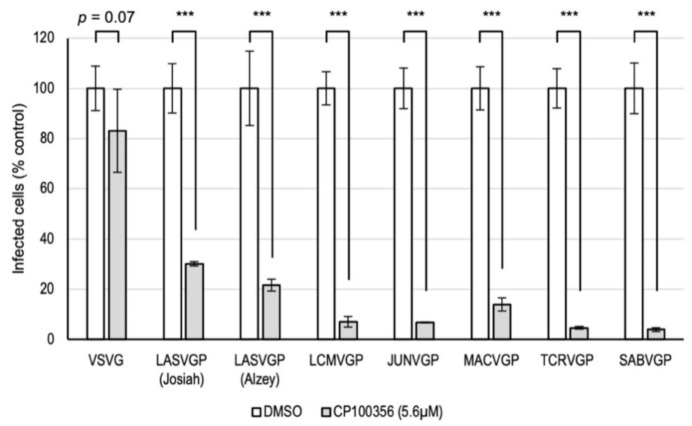
Broad-spectrum inhibitory effects of CP100356 against different mammarenaviruses. Vero cells treated with CP100356 or DMSO were infected with pseudotyped viruses harboring vesicular stomatitis virus glycoprotein (VSVG), LASV glycoprotein (LASVGP) (strain Josiah), LASVGP (strain Alzey), LCMVGP, JUNVGP, MACVGP, TCRVGP, or SABVGP, respectively. Infection was evaluated at 20 h post-infection by counting the number of cells expressing green fluorescent protein relative to the number of nuclei in the wells. Welch’s t-test was used to compare the two groups, and *p* values are indicated; *** *p* < 0.001. The values indicate means ± standard deviation of representative experiments in biological triplicates from two independent experiments.

## Data Availability

Not applicable.

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
