# Peer review of "CP100356 Hydrochloride, a P-Glycoprotein Inhibitor, Inhibits Lassa Virus Entry: Implication of a Candidate Pan-Mammarenavirus Entry Inhibitor"

_viruses, 2021, doi:10.3390/v13091763_

Round 1

Reviewer 1 Report

The manuscript by Takenaga and co-authors reports the findings of a small compound chemical library screen in hope to find entry inhibitors for Lassa virus. The authors identified a P-glycoprotein inhibitor, CP100356, as an efficient inhibitor of LASV and LCMV infection in vitro. The authors describe the initial screening very clearly, and especially Fig. 1 is a very overview of the procedure. Then the authors went on to test some of the promising inhibitors with real arenaviruses and attempted to identify the inhibition mechanisms, ruling e.g. out direct inhibition of receptor binding. I am not sure if Vero E6 is the most relevant cell line to perform the screen, perhaps some other cell line (e.g. A549 or BHK21) would have provided more hits. Although the “plain VSV” control is a good one to have, it seems possible that some of the inhibitors ruled out using such control could have been interesting as more general/broad range inhibitors of virus fusion (as also seems to be the case for CP100356). The results (Figure 5) show CP100356 to be an efficient general inhibitor of mammarenaviruses, however, this does not mean that the inhibitor would be a pan-arenavirus inhibitor. To make such a claim the authors would need to perform the experiments with hartmani-, reptarena- and antennavirus GPs. The title need to be revised accordingly i.e. pan-arenavirus => pan-mammarenavirus. The only way the authors actually acknowledge the existence of these viruses is by using mammarenavirus instead of arenavirus.

Some specific comments to improve the manuscript:

Line 24, currently used twice, please revise.

Line 25, how did the authors know at this point that these compounds have potential antiviral activity? Do the authors rather mean that they “screened a library of 2480 compounds for their potential…”?

Line 42, serogroup or complex instead of group?

Line 46, remove also?

Line 47, this refers to the New World arenas, right? Please revise to make it clearer.

Line 51, remove However.

Line 55, remove also

Line 88, from the New World serogroup?

Figure 2, do the authors mean (I could not easily extract the info from mat&met) that they collected the supernatant from the cultures at the indicated time points and then determined the amount of virus in the supernatants by some assay? Or do did the authors just quantify/enumerate the number of infected cells? If the authors just calculated the number of infected cells, then the y-axis cannot be PFU/ml or any infectious units/ml. I would rather present it as % against the negative control i.e. the number gained without the inhibitor. If the titers were actually defined, then please revise to give a clearer idea of the methodology.

Line 270, reduced the number of infected cells rather than the titer?

Figure 3A, although not statistically significant (due to the large error bars in the control?), it seems that the chemical could have slight inhibitory effect to the receptor binding. Given that the result is not that clear, it would perhaps have been better to show a broader range of inhibitor concentrations in this assay. Why not use the same concentration range as in Fig. 1, that should provide a clear results.

Line 323 and elsewhere, the authors used Vero E6 cells and not Vero, correct? Please revise.

Figure 4B, the unit should not in my opinion be titer but rather something else e.g. number of infected cells or something.

-line 356, please revise to pan-mammarenavirus or perform and include experiments with hartmani-, reptarena-, and antennavirus GPs.

Author Response

Reviewer #1

The manuscript by Takenaga and co-authors reports the findings of a small compound chemical library screen in hope to find entry inhibitors for Lassa virus. The authors identified a P-glycoprotein inhibitor, CP100356, as an efficient inhibitor of LASV and LCMV infection in vitro. The authors describe the initial screening very clearly, and especially Fig. 1 is a very overview of the procedure. Then the authors went on to test some of the promising inhibitors with real arenaviruses and attempted to identify the inhibition mechanisms, ruling e.g. out direct inhibition of receptor binding. I am not sure if Vero E6 is the most relevant cell line to perform the screen, perhaps some other cell line (e.g. A549 or BHK21) would have provided more hits. Although the “plain VSV” control is a good one to have, it seems possible that some of the inhibitors ruled out using such control could have been interesting as more general/broad range inhibitors of virus fusion (as also seems to be the case for CP100356). The results (Figure 5) show CP100356 to be an efficient general inhibitor of mammarenaviruses, however, this does not mean that the inhibitor would be a pan-arenavirus inhibitor. To make such a claim the authors would need to perform the experiments with hartmani-, reptarena- and antennavirus GPs. The title need to be revised accordingly i.e. pan-arenavirus => pan-mammarenavirus. The only way the authors actually acknowledge the existence of these viruses is by using mammarenavirus instead of arenavirus.

We are very grateful to the Reviewer for the evaluation of our manuscript and constructive comments. As suggested by this reviewer, we revised the title to “CP100356 hydrochloride, a P-glycoprotein inhibitor, inhibits Lassa virus entry: Implication of a candidate pan-mammarenavirus entry inhibitor” in line 3.

Some specific comments to improve the manuscript:

Line 24, currently used twice, please revise.

We have deleted the second “currently” in line 24.

Line 25, how did the authors know at this point that these compounds have potential antiviral activity? Do the authors rather mean that they “screened a library of 2480 compounds for their potential…”?

As suggested by the reviewer, we have revised the sentence to “We screened 2,480 small compounds for their potential antiviral activity using pseudo-typed vesicular stomatitis virus harboring the LASV glycoprotein” in line 25.

Line 42, serogroup or complex instead of group?

We have revised the term to “The Old World serogroup” in line 44.

Line 46, remove also?

We have deleted the word “also” in line 48.

Line 47, this refers to the New World arenas, right? Please revise to make it clearer.

In response to the reviewer’s comment, we have revised the sentence to “these highly pathogenic mammarenaviruses including LASV are classified as Category A bioterrorism agents” in lines 49-50.

Line 51, remove However.

We have deleted the word “However” in line 53.

Line 55, remove also

We have deleted the word “also” in line 57.

Line 88, from the New World serogroup?

We have revised the term to “New World serogroup” in line 91.

Figure 2, do the authors mean (I could not easily extract the info from mat&met) that they collected the supernatant from the cultures at the indicated time points and then determined the amount of virus in the supernatants by some assay? Or do did the authors just quantify/enumerate the number of infected cells? If the authors just calculated the number of infected cells, then the y-axis cannot be PFU/ml or any infectious units/ml. I would rather present it as % against the negative control i.e. the number gained without the inhibitor. If the titers were actually defined, then please revise to give a clearer idea of the methodology.

We apologize for the vague description. In the experiments shown in Figure 2, we collected the cell culture supernatants from virus-infected cells at indicated time points and determined the virus titers. To make it clearer, we have revised the sentence to “The supernatants were collected at indicated time points, and the respective virus titers were determined.” in line 257.

Line 270, reduced the number of infected cells rather than the titer?

We apologize for the vague description. In this experiment, we evaluated the impact of CP100356 treatment on LASV replication by determining the virus titers by TCID50 assay. To avoid misunderstanding, we provided additional information ‘Then, the virus titers were determined by TCID50 assay’ in line 278, and revised the term “infection” to “replication” in line 279.

Figure 3A, although not statistically significant (due to the large error bars in the control?), it seems that the chemical could have slight inhibitory effect to the receptor binding. Given that the result is not that clear, it would perhaps have been better to show a broader range of inhibitor concentrations in this assay. Why not use the same concentration range as in Fig. 1, that should provide a clear results.

We appreciate this reviewer’s constructive comment. Because the CC50 value of CP100356 was approximately 21 μM as shown in Figure 2A, we treated the cells with the highest concentration without showing significant cytotoxicity, i.e. 20 μM, for this assay. As shown in Figure 2B, treatment of 20 uM CP100356 strongly inhibited VSV-LASVGP replication. Thus, we consider that the experimental condition is appropriate for this experiment.

Line 323 and elsewhere, the authors used Vero E6 cells and not Vero, correct? Please revise.

Throughout this study, we used Vero cells, except for the cell-based membrane fusion assay (HEK-293T cells) and LASV infection experiments in BSL-4 (Vero E6 cells). Therefore, the provided information regarding the use of specific cell types is correct.

Figure 4B, the unit should not in my opinion be titer but rather something else e.g. number of infected cells or something.

We apologize for the vague description. In this experiment, we evaluated the impact of P-gp downregulation on respective virus replication. To avoid misunderstanding, we have revised the term “infection” to “replication” in lines 336 and 350.

line 356, please revise to pan-mammarenavirus or perform and include experiments with hartmani-, reptarena-, and antennavirus GPs.

We have revised the term to “a pan-mammarenavirus” in line 369.

Reviewer 2 Report

This is a well presented study by Takenaga and collaborators, describing the antiviral activity of the P-glycoprotein inhibitor CP100356 against several arenaviruses. This study is of importance considering the lack of antiviral strategies against arenaviruses. The authors propose that CP100356 inhibits viral entry by blocking membrane fusion, probably acting on a cellular process required for fusion.

While the results of this study are interesting, several aspects need to be addressed to improve the overall quality of the study and to convince about the specificity of the drug.

Specific comments:

  1. In the original screen Figure 1, 10 compounds decreasing VSV replication by less than two folds were further selected for testing against LCMV. It is stated that a concentration of 10 μM was used in the screen. However, at this concentration, CP100356 is likely to inhibit VSV replication by more than two folds, as described in Fig. 2D. Authors should comment on this point.
  2. In Fig.2, VSV is used as a negative control but it replicates at 4 log higher than VSV-LASVGP. Thus, VSV-LASVGP is attenuated compared to VSV and the two viruses may not have the same growth kinetics. VSV also replicates much faster than LCMV and LASV and this may affect the readout. Another VSV-XGP virus, encoding the GP of an unrelated virus (EBOV?) and presenting the same kinetics than VSV-LASVGP should be used a negative control. The same is true for Fig. 5. 
  3. Fig. 3B, it is difficult to assess the amount of fusion from the pictures. Authors should quantify the amount of fusion.
  4. Fig. 3C, authors should explain why they have used a high concentration of CP100356 (20 μM, about the CC50 according to Fig. 2). Would CP100356 still provoke accumulation of virus particles in the intracellular vesicles at lower concentration?
  5. Fig. 4, authors downregulated expression of P-gp by siRNA silencing. The silencing efficiency is not great (quantification?) and the data on LCMV suggest a trend towards inhibition of the relative virus titer after silencing. Considering that these are the results of two independent experiments, one may wonder if a third experiment would render the differences significant for LCMV. Thus is not clear whether P-gp is important or not for entry in presence of the drug. A better silencing efficiency or a KO cell line for P-gp would be more appropriate to address this point. Authors may also consider other available drugs targeting P-GP, such as Tariquidar.

Author Response

Reviewer #2

This is a well presented study by Takenaga and collaborators, describing the antiviral activity of the P-glycoprotein inhibitor CP100356 against several arenaviruses. This study is of importance considering the lack of antiviral strategies against arenaviruses. The authors propose that CP100356 inhibits viral entry by blocking membrane fusion, probably acting on a cellular process required for fusion. While the results of this study are interesting, several aspects need to be addressed to improve the overall quality of the study and to convince about the specificity of the drug.

We are very grateful to the Reviewer for the positive evaluation of our manuscript. We hope that our responses detailed below are satisfactory.

Specific comments:

  1. In the original screen Figure 1, 10 compounds decreasing VSV replication by less than two folds were further selected for testing against LCMV. It is stated that a concentration of 10 μM was used in the screen. However, at this concentration, CP100356 is likely to inhibit VSV replication by more than two folds, as described in Fig. 2D. Authors should comment on this point.

Our initial screening experiments included many compounds. Thus, we performed the experiments with slightly simplified procedures (e.g., we omitted washing step with PBS after virus infection). This may explain the slightly different results shown in Figure 1 and Figure 2 in terms of the suppression of virus titers.

  1. In Fig.2, VSV is used as a negative control but it replicates at 4 log higher than VSV-LASVGP. Thus, VSV-LASVGP is attenuated compared to VSV and the two viruses may not have the same growth kinetics. VSV also replicates much faster than LCMV and LASV and this may affect the readout. Another VSV-XGP virus, encoding the GP of an unrelated virus (EBOV?) and presenting the same kinetics than VSV-LASVGP should be used a negative control. The same is true for Fig. 5. 

We appreciate this reviewer’s thoughtful suggestions. Unfortunately, a recombinant VSV-EBOVGP is not available in our laboratory. Permission for the use of this genetically modified virus would require several months. Given the observed differences for VSV-WT and VSV-LASV in our experiments, we think that VSV wild type represents an appropriate negative control, despite different growth kinetics.

  1. 3B, it is difficult to assess the amount of fusion from the pictures. Authors should quantify the amount of fusion.

We tried to quantify the inhibitory effect of CP100356 on cell-to-cell fusion from the fluorescence images. However, because fused cells showed weak GFP signals, it was technically difficult to trace the area of syncytium formation and we were not able to quantify the fusion activity. We hope that the images shown in Figure 3B are convincing, demonstrating that CP100356 inhibits LASVGP and LCMVGP-mediated cell-to-cell fusion.

  1. 3C, authors should explain why they have used a high concentration of CP100356 (20 μM, about the CC50 according to Fig. 2). Would CP100356 still provoke accumulation of virus particles in the intracellular vesicles at lower concentration?

For electron microscopy analysis, ideally high MOI must be used to robustly visualize the virus particles within the cells following virus entry, while the use of low MOI is sufficient for virus titration in multicycle replication experiments (Figure 1-2). For that reason, we treated the cells with the highest concentration without showing significant cytotoxicity, i.e. 20 μM, to clearly evaluate the impact of CP100356 on virus entry. Although we have not performed electron microscopy analysis with a lower concentration, our virological data (Figure 2 and 3) strongly suggests that CP100356 causes virus particle accumulation in intracellular vesicles even at lower concentrations.

  1. 4, authors downregulated expression of P-gp by siRNA silencing. The silencing efficiency is not great (quantification?) and the data on LCMV suggest a trend towards inhibition of the relative virus titer after silencing. Considering that these are the results of two independent experiments, one may wonder if a third experiment would render the differences significant for LCMV. Thus is not clear whether P-gp is important or not for entry in presence of the drug. A better silencing efficiency or a KO cell line for P-gp would be more appropriate to address this point. Authors may also consider other available drugs targeting P-GP, such as Tariquidar.

As this reviewer pointed out, P-gp downregulation might affect LCMV entry based on the result in Figure 4B, while it did not affect LASV entry. Although the reduction on the LCMV replication was not significant, we agree with this reviewer that its impact on LCMV replication cannot be excluded. Thus, we have deleted ‘LCMV’ from the sentence and revised to “The efficiency of VSV-LASVGP and VSV replication in the knockdown cells was comparable with that in the control cells (Figure 4B), suggesting that the inhibitory effect of CP100356 on LASVGP-mediated membrane fusion is independent of P-gp.” in lines 338-340.

Reviewer 3 Report

Review Report MDPI-Journal Viruses (ISSN 1999-4915) Manuscript ID viruses-1329617

CP100356 hydrochloride, a P-glycoprotein inhibitor, inhibits Lassa virus entry: Implication of a candidate pan-arenavirus entry inhibitor by Toru Takenaga , Zihan Zhang , Yukiko Muramoto , Sarah Katharina Fehling , Ai Hirabayashi , Yuki Takamatsu , Junichi Kajikawa , Sho Miyamoto , Masahiro Nakano , Shuzo Urata , Allison Groseth , Thomas Strecker , Takeshi Noda *

Takenaga and coworkers analyzed a chemical library consisting of 2,480 small compounds   from a library to identify those inhibitors of the entry of Lassavirus in Vero cells using a pseudo-typed vesicular stomatitis virus (VSV) harboring LASVGP (VSV-LASVGP). After an elegant scaling down procedure (Fig.1) the authors end up with two compounds without significant cytotoxicity , CP100356 hydrochloride  (Fig. 1B) and MCOPPB trihydrochlo-236 ride hydrate. Only CP100356 was selected for further investigation (Fig.1B). The authors studied cell viability, and influence on virus titers dependent of inhibitor concentrations of pseudo-typed Lassa-GP-VSV, Lassa virus, and LCMV (Fig.2). Cell fusion at low pH (control without lowering the pH). Virus particles per endosome (Fig.3). The CP100356 indeed shows a clear broad inhibitory effect on cells infected by two strains of Lassa viruses and by other arenaviruses. The antiviral mechanism of CP100356 seems to be indirect and CP100356 is suggested as a promising basis for development of anti-arenaviral agents.

Please, give more necessary information in the introduction:

It is obligatory to give sufficient information to colleagues for understanding and appraisal of your work. Otherwise, it is hard to guess what background you start from, what is your motivation and whom you like to address. It is not acceptable that colleagues must collect quite a number of articles from literature or ask Wikipedia to follow your text. Your work should be attractively presented to scientific readers.  

1st  P-Glycoprotein is absolutely necessary to be introduced, for instances, as multidrug resistance protein 1 (MDR1) or ATP-binding cassette sub-family B member 1 (ABCB1) or cluster of differentiation 243 (CD243) is an important protein of the cell membrane that pumps many foreign substances out of cells. Its functions etc... , also with some relevant references or 1 to 2 reviews, and not only mentioned in the Discussion,

2nd The inhibitors used in this article needs also to be characterized by chemical nomenclature, references etc.... Why is the MCOPPB abbreviated for 1-[1-(1-Methylcyclooctyl)-4-piperidinyl]-2-[(3R)-3-piperidinyl]-1H-benzimidazole trihydrochloride hydrate produces potent anxiolytic effects omitted from this study, whereas only CP100356 4-(3,4-Dihydro-6,7-dimethoxy-2(1H)-isoquinolinyl)-N-2[2-(3,4-dimethoxyphenyl)ethyl]-6,7-dimethoxy-2-quinazolinamine hydrochloride, is an inhibitor for P-glycoprotein 1.

3rd ...For screening, a total of 2,480 small compounds were provided by the Medical Research Support Center, Graduate School of Medicine, Kyoto University (page 3, lines 110, 111). This library should be closer described in more detail, only pharmaceutical compounds? which selected compounds? why this library not others, or was it just arbitrarily? what expectation?...

I missed necessary and fair citations for cleavage trimerization of Lassa glycoprotein (page 2; line 70: #17 as only reference is not adequate and needs more originated references (Lenz et al. 2001; Eichler et al. 2003; Schlie et al. 2010; for LCMV-GP2 only Eschli et al. 2006.

4th The syncytium formation figure 3 could be given more precisely by polykaryon indexes. The unusuable low pH4.5 for fusion should be referred to Klewitz et al. 2007 and Ref. Jae et al #19.

Author Response

Reviewer #3

Takenaga and coworkers analyzed a chemical library consisting of 2,480 small compounds from a library to identify those inhibitors of the entry of Lassavirus in Vero cells using a pseudo-typed vesicular stomatitis virus (VSV) harboring LASVGP (VSV-LASVGP). After an elegant scaling down procedure (Fig.1) the authors end up with two compounds without significant cytotoxicity, CP100356 hydrochloride (Fig. 1B) and MCOPPB trihydrochlo-236 ride hydrate. Only CP100356 was selected for further investigation (Fig.1B). The authors studied cell viability, and influence on virus titers dependent of inhibitor concentrations of pseudo-typed Lassa-GP-VSV, Lassa virus, and LCMV (Fig.2). Cell fusion at low pH (control without lowering the pH). Virus particles per endosome (Fig.3). The CP100356 indeed shows a clear broad inhibitory effect on cells infected by two strains of Lassa viruses and by other arenaviruses. The antiviral mechanism of CP100356 seems to be indirect and CP100356 is suggested as a promising basis for development of anti-arenaviral agents.

Please, give more necessary information in the introduction:

It is obligatory to give sufficient information to colleagues for understanding and appraisal of your work. Otherwise, it is hard to guess what background you start from, what is your motivation and whom you like to address. It is not acceptable that colleagues must collect quite a number of articles from literature or ask Wikipedia to follow your text. Your work should be attractively presented to scientific readers.  

We appreciate the helpful comments from this reviewer to improve our manuscript. As suggested, we have revised the introduction section described below.

1st  P-Glycoprotein is absolutely necessary to be introduced, for instances, as multidrug resistance protein 1 (MDR1) or ATP-binding cassette sub-family B member 1 (ABCB1) or cluster of differentiation 243 (CD243) is an important protein of the cell membrane that pumps many foreign substances out of cells. Its functions etc... , also with some relevant references or 1 to 2 reviews, and not only mentioned in the Discussion,

As suggested by the reviewer, we have added the description about p-glycoprotein in lines 91-94 referring to two papers.

#24 Robey, R.W.; Pluchino, K.M.; Hall, M.D.; Fojo, A.T.; Bates, S.E.; Gottesman, M.M. Revisiting the Role of ABC Transporters in Multidrug-Resistant Cancer. Nat. Rev. Cancer 2018, 18, 452–464, doi:10.1038/s41568-018-0005-8.

#25 Mollazadeh, S.; Sahebkar, A.; Hadizadeh, F.; Behravan, J.; Arabzadeh, S. Structural and Functional Aspects of P-Glycoprotein and Its Inhibitors. Life Sci. 2018, 214, 118–123, doi:10.1016/j.lfs.2018.10.048.

2nd The inhibitors used in this article needs also to be characterized by chemical nomenclature, references etc.... Why is the MCOPPB abbreviated for 1-[1-(1-Methylcyclooctyl)-4-piperidinyl]-2-[(3R)-3-piperidinyl]-1H-benzimidazole trihydrochloride hydrate produces potent anxiolytic effects omitted from this study, whereas only CP100356 4-(3,4-Dihydro-6,7-dimethoxy-2(1H)-isoquinolinyl)-N-2[2-(3,4-dimethoxyphenyl)ethyl]-6,7-dimethoxy-2-quinazolinamine hydrochloride, is an inhibitor for P-glycoprotein 1.

As suggested by the reviewer, we have added the compound names characterized by chemical nomenclature in lines 244-247. As already described in lines 248-251, because its target P-gp is highly expressed on epithelial cell surfaces in the liver, kidney, and gastrointestinal tract, some of which are important replication sites of LASV in vivo, we selected CP100356 for further study.

3rd ...For screening, a total of 2,480 small compounds were provided by the Medical Research Support Center, Graduate School of Medicine, Kyoto University (page 3, lines 110, 111). This library should be closer described in more detail, only pharmaceutical compounds? which selected compounds? why this library not others, or was it just arbitrarily? what expectation?...

The compound library of the Medical Research Support Center, Kyoto University, comprises 2,480 pharmacologically active and validated small compounds that are purchased from various commercial vendors. We have added the information in lines 210-212.

I missed necessary and fair citations for cleavage trimerization of Lassa glycoprotein (page 2; line 70: #17 as only reference is not adequate and needs more originated references (Lenz et al. 2001; Eichler et al. 2003; Schlie et al. 2010; for LCMV-GP2 only Eschli et al. 2006.

We appreciate the important comment by the reviewer. As suggested, we have added the 4 references below in line 72.

#18 Lenz, O.; ter Meulen, J.; Klenk, H.D.; Seidah, N.G.; Garten, W. The Lassa Virus Glycoprotein Precursor GP-C Is Proteolytically Processed by Subtilase SKI-1/S1P. Proc. Natl. Acad. Sci. U. S. A. 2001, 98, 12701–12705, doi:10.1073/pnas.221447598.

#19 Eichler, R.; Lenz, O.; Strecker, T.; Eickmann, M.; Klenk, H.-D.; Garten, W. Identification of Lassa Virus Glycoprotein Signal Peptide as a Trans-Acting Maturation Factor. EMBO Rep. 2003, 4, 1084–1088, doi:10.1038/sj.embor.embor7400002.

#20 Schlie, K.; Strecker, T.; Garten, W. Maturation Cleavage within the Ectodomain of Lassa Virus Glycoprotein Relies on Stabilization by the Cytoplasmic Tail. FEBS Lett. 2010, 584, 4379–4382, doi:10.1016/j.febslet.2010.09.032.

#21 Eschli, B.; Quirin, K.; Wepf, A.; Weber, J.; Zinkernagel, R.; Hengartner, H. Identification of an N-Terminal Trimeric Coiled-Coil Core within Arenavirus Glycoprotein 2 Permits Assignment to Class I Viral Fusion Proteins. J. Virol. 2006, 80, 5897–5907, doi:10.1128/JVI.00008-06.

4th The syncytium formation figure 3 could be given more precisely by polykaryon indexes. The unusuable low pH4.5 for fusion should be referred to Klewitz et al. 2007 and Ref. Jae et al #19.

In response to the reviewer’s comment, we re-analyzed the images with nuclear staining using Hoechst 33342 and found that the nuclear signals were specifically weak in fused cells compared to those in unfused cells as shown below. As we did not clearly observe multiple nuclei in GP-mediated fused cells, we believe that we do not need to present the data in figure 3. As for the unusable low pH procedure, we have added the references in lines 175.

#31 Klewitz, C.; Klenk, H.-D.; Ter Meulen, J. Amino Acids from Both N-Terminal Hydrophobic Regions of the Lassa Virus Envelope Glycoprotein GP-2 Are Critical for PH-Dependent Membrane Fusion and Infectivity. J. Gen. Virol. 2007, 88, 2320–2328, doi:10.1099/vir.0.82950-0.

#23 Jae, L.T.; Raaben, M.; Herbert, A.S.; Kuehne, A.I.; Wirchnianski, A.S.; Soh, T.K.; Stubbs, S.H.; Janssen, H.; Damme, M.; Saftig, P.; et al. Virus Entry. Lassa Virus Entry Requires a Trigger-Induced Receptor Switch. Science 2014, 344, 1506–1510, doi:10.1126/science.1252480.

Round 2

Reviewer 2 Report

Authors have tried to address my comments but I still have doubts about the specificity of the drug without the suggested experiments that were not performed. I also believe that the title of the manuscript should be modified as the drug does not inhibit "Lassa virus entry" but inhibits entry of a LASV-GP pseudotyped VSV.

Author Response

We are very grateful to the Reviewer for the valuable comments. I hope our responses are satisfactory.

Authors have tried to address my comments but I still have doubts about the specificity of the drug without the suggested experiments that were not performed.

Because other recombinant VSV viruses, such as a recombinant VSV-EBOVGP virus, are not available in our laboratory, we could not perform the suggested experiments, as described in the last response. However, as pointed out by the reviewer, we consider that the specificity and broad spectrum of CP100356 is an important point as a drug candidate. After we have a permission for the use of the genetically modified viruses (e.g., VSV-EBOVGP and VSV-MARVGP), we would like to investigate whether CP100356 also works against filoviruses in the next study.

I also believe that the title of the manuscript should be modified as the drug does not inhibit "Lassa virus entry" but inhibits entry of a LASV-GP pseudotyped VSV.

Because CP100356 strongly inhibited both VSV-LASVGP and authentic LASV replication, but only modestly inhibited VSV replication, we consider that CP100356 inhibits LASVGP-mediated virus entry. Thus, we believe that our title “CP100356 hydrochloride inhibits Lassa virus entry” is acceptable.
